# Laser-Generated Scholte Waves in Floating Microparticles

**DOI:** 10.3390/s23041776

**Published:** 2023-02-04

**Authors:** Abhishek Ranjan, Azeem Ahmad, Balpreet Singh Ahluwalia, Frank Melandsø

**Affiliations:** Department of Physics and Technology, UiT The Arctic University of Norway, 9037 Tromsø, Norway

**Keywords:** laser, ultrasound, microparticles, COMSOL, Scholte wave

## Abstract

This study aims to demonstrate the generation and detection of Scholte waves inside polystyrene microparticles. This was proven using both experimental analysis and COMSOL simulation. Microspheres of different sizes were excited optically with a pulsed laser (532 nm), and the acoustic signals were detected using a transducer (40 MHz). On analyzing the laser-generated ultrasound signals, the results obtained experimentally and from COMSOL are in close agreement both in the time and frequency domain. A simplified analysis of Scholte wave generation by laser irradiation for homogeneous, isotropic microspheres is presented. The theoretical wave velocity of the Scholte wave was calculated and found close to our experimental results. A representation of pressure wave motion showing the Scholte wave generation is presented at different times.

## 1. Introduction

Laser-generated ultrasound technology has been of great interest due to its application in the non-destructive evaluation of solid surfaces and thin films [1,2]. This technology is based on the photoacoustic effect where the sample is optically excited and ultrasonically detected [3,4]. The absorbing samples are irradiated by a pulsed laser, when the laser hits the sample; light is absorbed, which subsequently leads to heating. Heating further leads to thermoelastic expansion eventually giving rise to pressure waves. The pressure waves are finally detected by the transducer. The properties of the laser-generated ultrasonic waves depend strongly not only on the optical penetration [5], thermal diffusion [6], and elastic and geometrical features of the materials but also on the parameters of the exciting laser pulse, including the shape, the focus spot, and pulse width. Different kinds of ultrasonic waves can be excited such as longitudinal and transverse waves, surface acoustic waves, and lamb waves in the thermoelastic excitation regime [7]. Surface waves are often found in nature in a variety of forms, for instance, seismic waves along the Earth’s crust, water waves that follow along the air-water interface, and ultrasonic surface waves at different interfaces. In all these cases, a standard feature of surface acoustic waves (SAW) is that energy is localized near the surface, and the sound intensity decreases exponentially with propagation in the depth of the material [8,9]. In the bulk of an elastic material, surface acoustic waves depend on the elastic deformation happening between the constituent atoms. The longitudinal and the transverse waves, which travel in the bulk of the material, are independent traveling with different velocities, but these two modes are coupled on surface waves. The speed of surface waves also differs from the bulk waves because particles are less constrained at the free surface than in the interior of the material [10].

Longitudinal, transverse waves, surface acoustic waves, and lamb waves have been shown in thin plates [11]. JD Achenbach demonstrated that for a point focus illumination the surface wave pulse consists of a main pulse followed by a smaller pulse later [12]. The backscattered transfer function of microspheres and single cells was modelled using a Faran-Hickling solution, and a good agreement was found between the theoretically predicted and experimental results for polystyrene micro-sized particles using ultrasound microscopy techniques [13]. Micron-sized droplets loaded with nanoparticles have been investigated, and the power spectrum has been compared with the theoretical model and experimentally [14]. Laser-generated ultrasound models for additive manufacturing materials have been utilized analytically during the cooling process [15,16].

Laser-generated ultrasounds have been utilized for microparticles earlier and have been shown for several applications. For instance, microparticles using photoacoustics have been shown to be utilized in image-guided cancer therapy [17]. Cells were incubated with microparticles to determine the intracellular temperature using changes in the photoacoustic amplitude [18]. Xiaoxiang Gao et al. has demonstrated that eigen vibration information of light-absorbing microstructures can be obtained from photoacoustic amplitudes enabling noncontact evaluation of elastic properties [19]. Noncontact and non-invasive low-cycle fatigue characterization of homogenous, isotropic, elastic cylinders via photoacoustic eigen-spectrum analysis have been shown [20]. A.B. Matsko et al. demonstrated acoustic coupling between acoustic whispering gallery modes created by the surface acoustic waves [21]. SAWs have been used to find acoustic properties of the near-surface layer using the property that SAW travels parallel to the surface [2]. Micron-sized droplets of various sizes loaded with nanoparticles have been analysed with laser-generated ultrasound both experimentally and theoretically earlier. However, a study of a different kind of waves emitted inside microparticles together, with the study of eigen frequencies both experimentally and with a theoretical model for a solid microsphere, and with the detection of Scholte waves is missing so far in the literature to the best of our knowledge.

This study aims to investigate the existence of acoustic waves inside microparticles both experimentally and theoretically using laser irradiation. We report photoacoustic experiments of microparticles of varied sizes, having diameters of 20, 40, and 80 µm. This study develops a model using Beer–Lambert’s law, which includes the effects of finite width and the duration of the laser source, laser fluence, the material properties of the solid under investigation, and acoustic receivers. We present the modelling of the generation of laser-generated acoustic waves in COMSOL by coupling the heat diffusion equation and the momentum equation. We detect and prove the presence of Scholte waves on the interface of microspheres and water theoretically and experimentally. The velocity for the Scholte wave is calculated both theoretically and experimentally. We present an analysis of pressure waves originating at different times for microparticles and calculate the velocities of different waves emerging out. To our knowledge, this paper is the first demonstration of the detection of Scholte waves on the interface of microspheres and water theoretically and experimentally. We also compare the eigenfrequency locations in the frequency response of the signals received from microparticles both numerically and experimentally. The predicted eigenfrequency results match very well with the experimentally obtained eigenvalues.

## 2. Materials and Methods

### 2.1. Preparation of Agarose

To prepare the agarose, 0.1 g of agarose powder (Sigma Aldrich (Darmstadt, Germany, Type I, low EEO)) is dissolved with 10 mL (1% agarose) of distilled water and poured into a beaker. The solution is then heated on a heat plate at a temperature of around 80 °C. A magnetic stirrer is placed inside the solution to stir the solution for better solubility. The agarose solution beaker is heated slowly for around 30 min to avoid the loss of water content due to evaporation. The molten agarose solution can then be poured either into a mold or directly on the Petri dish.

### 2.2. Multi-Layered Embedded Sample

A small volume of around 2 μL of molten agarose was poured onto the Petri dish to fill its periphery. The microparticles (density of 1050 kg/cc) were then mixed with molten agarose. A drop of molten agarose mixed with microparticles was then poured upon the agarose in Petri dish. The whole sample was then solidified so that the microparticles were embedded into the agarose matrix and at the same time lifted from the bottom of the Petri dish. This protocol was repeated similarly for microparticles of all sizes. The objective of this sample preparation technique was to embed the microparticles inside the agarose matrix so that they were floating inside the agarose and are boundaryless.

This protocol was repeated for microparticles of sizes namely PS20 (Batch number- PS-FR-L2858, mean size-19.90 µm), PS40 (Batch number-PS-FR-Fi217, mean size-41.11 µm), and PS80 (Batch number-PS-FR-Fi269, mean size-78.92 µm). The microparticle samples were made up of polystyrene and were produced along emulsion polymerization routes. The applied dye responsible for absorption on these polystyrene microparticles was Sudan red type. This dye was distributed uniformly throughout the volume in a homogeneous way.

## 3. Experimental Setup

To investigate the photoacoustic response from the microparticles, a customized experimental setup was built up around an inverted microscope (Leica DMi8) with additional optical and acoustic components as shown in Figure 1b. A high-precision scanning stage (ASI MS-2000, Eugene, OR, USA) was set up on the microscope to generate photoacoustic images of the microparticle samples using mechanical raster scanning. The stages and transducer movements were controlled using a LabVIEW program. This program assured synchronization with the optical and acoustic components for the needed communication and data transfer with the time-critical digitizing implemented in FPGA hardware (National Instruments, Austin, TX, USA). An optical camera was also added in wide-field reflection mode to provide optical images of the investigated microparticles.

The optical components shown below the scanning stage in Figure 1b were assembled on an optical table to minimize the effects of mechanical vibrations. These components consist of a Q-switched mode-locked pulsed laser (Elforlight SPOT-20-200-532, Daventry, UK) having a wavelength of 532 nm, which yields pulses with widths down to 1.6 ns, repetition rates up to 10 kHz, and energies up to 20 μJ per pulse. An optical isolator was placed immediately after the laser to avoid unwanted back reflection into the laser cavity and allow one-way transmission of laser light. The laser pulse was also spatially filtered using a 50-μm diameter pinhole to remove different aberrations and produce a laser beam having a smooth Gaussian intensity profile. The spatially filtered beam was used as an input to the inverted microscope through an infinity port and focused by a Leica objective lens (NA: 0.3, depth of field: 10 μm, magnification-10×, working distance-11 mm) at the focal plane. The acoustic waves were detected 32 times and averaged to reduce noise in each measurement.

The acoustic instruments of the setup acting as the receiver of acoustic waves were located above the scanning stage. This included a focused ultrasound transducer (Olympus, Tokyo, Japan, Serial number-200637) with a center frequency of around 40 MHz, a focal length of 12.5 mm, and a f-number of 2. In order to overlap the field-of-view for both the optical and the acoustic fields, the ultrasound transducer and the laser were coaxially and confocally aligned before imaging. Subsequently, to obtain good signal integrity, the signal from the transducer was amplified and low-pass filtered with a customized pre-amplifier and then digitized with a high-speed 12-bit digitizer (NI-5772) at a 400 MHz sampling rate.

## 4. COMSOL Simulation

A simulation was performed in COMSOL Multiphysics (Version 6.0) to understand the experimental results from the microparticles. The numerical model was constructed to understand the interaction of light with microparticles, and the detection of the acoustic waves generated. COMSOL Multiphysics uses the finite element method (FEM) for various space- and time-dependent problems, which are usually expressed in partial differential equations (PDE). This modeling assumed that the laser light was monochromatic, collimated, and had very minimal refraction, reflection, or scattering within the material itself. Agarose has an acoustic impedance very close to the water and was therefore modeled as a fluid with water parameters in our simulation. The model assumed the laser energy was absorbed instantaneously (infinitesimally narrow laser pulse) so that the pressure wave did not propagate prior to the start of the simulation. The laser intensity was kept below the ablation threshold, i.e., the acoustic generation process was under a thermoelastic regime. Additionally, 2-D asymmetric geometry was used in the model, with the regions specified as shown in Figure 2.

The laser beam was modelled as a Gaussian pulse, and it was assumed that the incident laser light intensity follows a Gaussian distribution with respect to distance from the origin when it enters another medium such as in our case microparticles, it remained Gaussian distribution. The pulse duration and beam diameter of the laser was kept as specified in Table 1. The microparticles absorb the laser light and produce acoustic waves and to model the absorption of laser light, Beer–Lambert’s law was used. Beer–Lambert’s Law denotes a relationship between the attenuation of light through a substance and its properties. According to this law, if the light hits a semi-transparent material, absorption of energy will take place assuming monochromatic and collimated light. The law can be described in its differential form as to:(1)$$\frac{\partial I}{\partial z}=\alpha \left(T\right)\;I\;$$
where I is the light intensity, z is the coordinate along the beam direction, T is the temperature, and $\alpha \left(T\right)$ is the absorption coefficient, which, in the general case, can be dependent on the temperature of the material. This law was implemented by adding the General form PDE interface. The way to further execute this in COMSOL is to solve the heat equation in the model and to relate them to the elastic wave equation. However, we assumed it to be temperature-independent since the temperature increase was small in our case, or $\alpha \left.\left(T\right.\right)=\alpha_{0}$. The heat diffusion equation describes about the distribution of heat or temperature variation over time inside the microparticle. It can be expressed as
(2)$${\rho C}_{p}\frac{\partial T}{\partial t}=\nabla \left(k\nabla T\right)+Q$$
where T is temperature, t is time, ρ is the mass density, $C_{p}$ is the heat capacity with constant pressure, ∇ is the del operator, k denotes the thermal conductivity, and Q = α(T)I represent the rate at which energy is generated per unit volume of the medium from an external heat source which equals to (laser in our case). If we assume thermal confinement, the diffusion from the heat flux term can be set to zero yielding.
(3)$${\rho C}_{p}\frac{\partial T}{\partial t}=Q$$

In photoacoustic (PA) imaging, thermal confinement refers to the process of confining the heat generated by the absorption of laser energy to a small region within a sample. This is typically achieved by using laser pulses that are short in duration (on the order of nanoseconds or picoseconds) and focused on a small spot size. The resulting rapid temperature rise and subsequent expansion of the heated material lead to the generation of ultrasound waves, which can be detected and used to create an image of the absorption properties of the sample.

Thermal confinement is important in PA imaging because it allows for the selective excitation of specific regions within a sample, which can be used to improve the contrast and resolution of the resulting images. It is also important for minimizing damage to the sample and for providing accurate measurements. There are several factors that can affect the extent of thermal confinement in PA imaging, including the absorption coefficient of the sample, the pulse duration and intensity of the laser, and the size and shape of the focus spot. Techniques such as pulse shaping and spatial filtering can be used to improve the thermal confinement and the overall performance of the PA imaging system, which is done in our experimental setup.

For effective PA signal generation, the laser pulse duration is typically several nanoseconds, which is less than both the thermal and stress confinement times. Thermal confinement implies that thermal diffusion during laser illumination can be ignored [22].
τ < τ_thres_= (d_c_^2^)/(4D_T_)(4)

Here, τ is the thermal confinement threshold, d_c_ is the desired spatial resolution, and D_T_ is the thermal diffusivity. Here, τ = 1.25 ns, d_c_ = 0.9 µm, D_T_ = 0.10204 µm, τ_thres_ = 1.98 µs. We can see that our laser pulse duration in(ns) is much less than the thermal confinement factor in (µs) which assures thermal confinement. For the COMSOL simulations included in this manuscript, we have solved the full heat equation given by Equation (2). We have also solved solutions for PS80 using Equation (3) where the heat flux is omitted, yielding no visible difference from comparing the received pulses. This COMSOL result therefore supports the analytically estimated timescales, suggesting that the heat flux can be neglected. Misael Ruiz-Veloz et al. have shown the case of solid samples with strong absorption properties, which are out of the validity range of the confinement conditions. It was demonstrated that by considering the thermal correction for the acoustic source, the 1-D wave equation may be a valid approximation for the generation and with thermal confinement and later proven by both theoretical modelling and experimental demonstrations [23].

The momentum equation i.e., Newton’s second law of motion in tensor form was solved for a linear elastic material where **u** is the displacement and **σ** is the stress tensor. This equation can be expressed as:(5)$$\rho \;\frac{\partial^{2}u\;}{\partial t^{2}}=\nabla \cdot \sigma$$
where the temperature change is affecting the elastic equation through the stress tensor **σ**. For an isotropic and linear elastic material, the components of this tensor can in the Voigt notation be expressed as: (6)$$\left[\begin{array}{c} \sigma_{11}\\ \sigma_{22}\\ \begin{array}{c} \sigma_{33}\\ \sigma_{23}\\ \begin{array}{c} \sigma_{13}\\ \sigma_{12} \end{array} \end{array} \end{array}\right]=\frac{E}{\left(1+v\right)\left(1-2v\right)}\left[\begin{array}{cccccc} 1-v & v & v & 0 & 0 & 0\\ v & 1-v & v & 0 & 0 & 0\\ v & v & 1-v & 0 & 0 & 0\\ 0 & 0 & 0 & \frac{\left(1-2v\right)}{2} & \frac{\left(1-2v\right)}{2} & 0\\ 0 & 0 & 0 & 0 & & \\ 0 & 0 & 0 & 0 & 0 & \frac{\left(1-2v\right)}{2} \end{array}\right]-\frac{{E\alpha }_{T}∆T}{1-2v}\;\left[\begin{array}{c} 1\\ \begin{array}{c} 1\\ \begin{array}{c} 1\\ 0\\ 0 \end{array} \end{array}\\ 0 \end{array}\right]$$
where E represents Young’s modulus, v is Poisson’s ratio, $\alpha_{T}$ is the coefficient of thermal expansion, and ∆T is the increase in temperature [24]. We also added the additional contribution to the diagonal of the stress tensor to compensate for the increased temperature. The units for every term specified in the equations above are in SI units. Different boundary conditions were chosen for different boundaries inside the model. Some of the laser light is reflected at the dielectric interface, leading in a reduction of light intensity at the material’s incident surface. This condition is implemented with a Dirichlet boundary condition. The zero-flux boundary condition was applied to the face opposite to the incident face to allow any light reaching that boundary to leave the domain. Most other boundaries were chosen to be default thermal insulation, fit for implementing the symmetry of the temperature field. The heat flux boundary condition is implemented in a heat transfer model at the interface of microparticle and water, which prescribes the heat per unit area flowing into (or out of) the model across the boundary. A spherical radiation boundary condition was implemented on the outer boundary to allow waves close to the spherical to leave with minimal reflections. In the table given below, we present a summary of the important parameters used in our model. Table 1 contains the details about the parameters related to the material and elastic properties of polystyrene microparticles used in the model. These parameters are specified as per the datasheet of the polystyrene manufacturer (microParticles, GmbH, Berlin, Germany).

An ultrasonic transducer was simulated in the model to receive the PA signals, which consisted of a piezoelectric polymer and a backing material. A thin film of Polyvinylidene Fluoride (PVDF) was implemented as a piezoelectric material similar to the transducer used in experimental setup. The ends of the piezoelectric layers were made grounded. The model was eventually solved in COMSOL using the transient response using a finite-size custom mesh size.

It was observed that by changing the pulse width resulted in a different frequency spectrum for the photoacoustic signal, which can also be proven experimentally by a slight change in the focus. This confirms that the pulse width of the laser beam changes after it interacts with the microparticle and inside the microparticles. The absorption coefficient is crucial as well because the photoacoustic signal is proportional to the absorption properties of the microparticle. It was observed in the modelling that the amplitude of the signal decreased with low absorption coefficients. The elastic property of microspheres such as mass density, Young’s modulus, and Poisson’s ratio of polystyrene microspheres also affected the photoacoustic signals in terms of the location of eigenfrequency.

## 5. Results and Discussion

The microparticles were embedded in an agarose matrix to simulate floating conditions. From comparing time series from floating and non-floating samples where the particles were in physical contact or in close vicinity of the Petri dish’s bottom, the floating samples typically yielded more long-lasting oscillations after laser excitations. This observation is strongly correlated to waves propagating at the particle-water interface, and the reduced attenuation of these waves when the microspheres are lifted-up from the petri dish. It was also observed that the proposed agarose method gave clear and repeatable signals for the used 532 nm laser, with an acceptable signal-to-noise ratio for all particle sizes. Figure 3a–c shows the photoacoustic images of different particles. It can be confirmed that the shape of the microparticles is spherical. For PS40 and PS80, shown in Figure 3b,c, respectively, the rings inside the microparticle can be observed, and in the center of the microparticle lies the core, which is surrounded by the shell. Studies have shown that the structure inside the microparticle depends on the method of their preparation [25,26,27]. The area shown in Figure 3 is 0.3 mm × 0.3 mm in *x*- and *y*-direction with a pixel size of 2 µm.

The laser-generated ultrasound signals from microparticles (PS20, PS40, and PS80) were obtained as shown in Figure 3d–f. The experiments were carried out several times for repeatability and verification. The photoacoustic signals are derived from the three center pixels of the microsphere and plotted on the same graph. It can be observed that the three signals are approximately similar in trend and amplitude. A Tukey window was applied on the signals in such a manner that removes the reflection coming from the bottom of Petri dish. After the laser pulse illumination, the microspheres absorb the incoming optical energy; this optical energy gets converted to heat. Subsequently, this leads to thermoelastic expansion, finally emitting PA waves to the surrounding media. The PA signals arising from the microparticles have a short period and high intensity, which corresponds to the rapid thermoelastic expansion of a microsphere induced by short-pulsed laser illumination. Previous studies have shown that this high-intensity signal coming from microparticles depends on the optical absorption coefficient [28] and the size of the absorbing particles [29,30,31]. It also reveals that even after the laser illumination is over, a periodic wave with a gradually decaying magnitude exists after a certain interval of time. It is observed from PA signals in the time-domain that the waves after the peak intensities repeat after a certain interval of time, which indicates that the microparticles vibrate even after narrow laser illumination is over. Upon further investigation, it can be clearly identified that the oscillation of signals follows a pattern of reoccurrence in time-domain which is a representative of different types of acoustic waves present inside the microparticles. The presence of different waves can be detected by calculation of their respective velocities. The different velocities match experimentally and theoretically in this work. The speeds of the longitudinal wave and shear wave, calculated here from our model, are approximately 2310 m/s and 1110 m/s, which is close to the known values [32].

We also performed scanning acoustic (ultrasound imaging) experiments with the microparticles for the same sample used in the PAM experiments. It is clearly observed that the resolution of the PAM image is better than the ultrasound image, as we can resolve the microparticles which are close to each other. The distance separation between the microparticle and the bottom of the Petri dish was calculated based on the signals received using time of flight method. The lifting height from the bottom of the Petri dish was calculated around 1.15 mm for PS40 and PS80 microparticles, respectively. The signals from the PS20 microparticle was not seen due to the low resolution of the transducer (f = 40 MHz) in ultrasound imaging. The following Figure 4 demonstrates the SAM signals in the time-domain and frequency-domain.

Eigenvalues can be defined as the natural frequencies of a system at which the system tends to vibrate without any driving or damping force. The microsphere can deform into a corresponding shape, also known as the eigenmode when vibrating at a certain eigenfrequency. The frequency responses of the microparticles were also numerically modeled in COMSOL, and they are in good agreement with the experimental results. Xiaoxiang Gao et al. has shown eigenvalue results for polystyrene microparticles having same elastic parameters (Young’s modulus and Poisson’s ratio) is in close proximity to the predicted eigenvalues for our PS20, thus confirming the validity of our results [20]. The eigenfrequencies depend inversely on the diameter of the microparticles. It is also observed from Figure 5g–i that number of eigenfrequencies increases with an increase in the particle’s diameter. On analysis, the experimental and the predicted eigenfrequency match in close correspondence to each other. We can compare the theoretically predicted and experimental eigenfrequency occurring for different microparticle sizes in Table 2.

On comparison of the frequency spectrum obtained experimentally and from the modelling, the peak eigenfrequency in the case of the 40 µm sample is lying almost at the same location in both cases as evident in Figure 3h and Figure 6e. The shape and the trend of the curves are also matching quite well in the case of signals from PS40 microparticles. We also obtain a good similarity in the trend and shape of the frequency spectrum in the experimental and simulated curves for PS80. We observe that the simulation holds in agreement with the experimental results for microparticles of larger sizes such as PS80 and PS40.

From the spectra shown in Figure 5d–f, we notice a quite good agreement for the PS40 and PS80 particles in terms of locations of the peak frequencies. These locations also match well with eigenfrequencies derived from the frequency responses shown in Figure 5h,i. For the PS20 particle, on the other hand, the COMSOL results predicts two peaks within the computed frequency band (0 to 70 MHz), while the experimental result shows only one. There might be several factors that contribute to the observed differences. One important factor is the amplifier used in the experiments after the transducer. This amplifier has an integrated analog band-pass filter with lower and upper cut-off frequencies around 10 MHz and 55 MHz, respectively. The experimental signals will therefore have a much stronger attenuation outside this band than the COMSOL signals where no analog filter is included yielding the natural frequency response from the particle scattering and the simplified transducer model. We notice that the second peak around 50 MHz in Figure 5d is relatively close to the upper cut-off. The filter is therefore expected to attenuate this eigenfrequency severely and could make it difficult to observe it as a peak value. Another factor that may alter the experimental spectra is the agarose gel used in the experiment. This gel is simulated as non-viscous fluid in COMSOL, meaning that the viscosity caused by gel-polymer coupling is not included. Since the particles’ area-to-volume ratios are largest for the smallest particles, the viscosity effects are consistent with the observed largest deviation for PS20, and the broadening of the experimental peak shown in Figure 5d.

Other factors that might contribute to deviation between experimental and simulated data are, for example, spatial anomalies from microparticles not exactly spherical in shape. Uncertainties in the elastic polystyrene parameters will also influence the eigenvalue locations. We have tried to input the elastic parameter values of polystyrene microparticles as close as possible in our model. The absorption coefficient of the microparticle is uncertain and not specified by the manufacturers. Although we have assumed agarose as water since the acoustic impedance of the agarose is close to water, agarose can cause some damping [6]. So, it is exceedingly difficult to get the exact trend as predicted, but we obtained a close match experimentally to the predicted results in the simulation. On analysing the Hilbert transform of the experimental signals with a phase shift of $\frac{\pi }{2}$ degrees to the original signal, a decrease of amplitude with the increase in the size of the microparticles can be observed and a periodic repetition after a certain interval. We observe a pattern that in the case of PS40, it is around 0.07 ± 0.2 µs, and in the case of PS80, µm, it is 0.14 ± 0.2 µs. This is theoretically correct as it takes double the time when the diameter of the microsphere is doubled. The experimental signal data of PS80 is noisy after t = 2 µs, so it can be difficult to differentiate the noise from the surface wave feature. It has been established that on a point focus illumination using a pulsed laser, the surface wave pulse consists of a principal wave followed by smaller pulses in previous works [1,13]. This confirms the presence of surface wave detection in our laser-generated ultrasound experiments.

From Figure 6,we notice that the pressure waves arising from microspheres at different time intervals in the simulation. In Figure 6a, the waves are seen when they are excited at the bottom, and with time, we can see that they are moving from the bottom to the top, as evident from Figure 6a–c. The surface wave is here converted to a longitudinal wave before getting detected by the transducer and this can be seen in Figure 6e. It can be observed that the Scholte wave is repeated evidently from Figure 6e–h. The first wave emerging is the longitudinal wave traveling along the diameter of the microparticle. The waves generated afterward take a longer time to travel as they travel along the surface covering the circumference of the microspheres. If we refer to the figure shown below, over several orbits, the amplitude decreases, the wave spread outs, and the spreading out is not very extensive, which is a clear indication of weak dispersion taking place. Since the dispersion is weak, we can still use frequency independent velocity for the estimation of bouncing time around the microspheres.

The Scholte wave velocity is calculated around 897 m/s for PS80 as per experimental results. This velocity was calculated experimentally by dividing the distance travelled by the repetition time interval (approximately 0.14 µs). This speed is slower than longitudinal and shear wave because of the wave dispersion. So, this can be recognized as the surface wave on microparticles surface. According to the definition, the Rayleigh waves are the ones that assume a vacuum surrounding the medium through which they pass [33]. As the microparticles embedded in agarose are surrounded by the water on all sides, the wave identified here is known as a Scholte wave due to its existence on the solid and liquid interface. The theoretical calculation of Scholte wave velocity was performed using the bisection method where the zero point for the dispersive function shown in the equation below was calculated [34]:(7)$${\mathrm{rx}}^{2\;}\sqrt{1-\beta x}+(2-x^{2\;})\;\sqrt{1-\beta^{*}x}-4\sqrt{1-x}\sqrt{1-\beta x}\sqrt{1-\beta^{*}x=0}$$
where $x=\frac{V_{SC}^{2}}{V_{S}^{2}}$, β = $\frac{V_{S}^{2}}{V_{P}^{2}}$, $\beta^{*}$ = $\frac{V_{S}^{2}}{V_{w}^{2}}$, r = $\frac{\;\rho \;}{\;\rho *}.$

For these expressions, $V_{P}$ and $V_{S}$ denotes the P-wave and S-wave microparticle velocities, ρ and ρ* the mass densities of water and the microparticle, and $V_{W}$ the acoustic velocity of water, respectively. The theoretically calculated value for the Scholte wave velocity ($V_{SC}$) as per Equation (7) was 860 m/s, which is calculated for flat surface geometry; however, in the case of the spherical geometry wave the velocity will be higher. This prediction matches very close to our experimentally calculated velocity, hence the presence of an interface wave, i.e., Scholte wave. The numerical computation we have done suggests that there is a weak dispersion. We have considered a weak dispersion, which might be a cause for the difference in the theoretical and experimental calculated velocity.

## 6. Summary and Conclusions

In this paper, we have studied laser-excited acoustic waves in microspheres using a pulsed laser. Our hypothesis was that the laser would initially excite both waves propagating inside in the microparticles (denoted as bulk waves) and waves propagating at the particle-water interface (Scholte waves). The COMSOL simulations have, to a large extent, confirmed this hypothesis for the PS40 and PS80 microspheres. For these spheres, the bulk waves typically dominate the initial phase after excitation, and Scholte waves are therefore observed only after some initial time. The simulations also show that the situation reverses as time increases, meaning that the wave activity will eventually be dominated by long-lasting Scholte wavelets orbiting the microspheres. Due to this behavior, we believe that time windowing of the data can be a possible approach for separating the contribution from the different wave types.

The theoretically calculated velocities for the Scholte wave agree quite well to the experimental findings. Pressure waves with evidence through the visualization of wave propagation at different times are explained. The experimentally generated and the numerically calculated acoustic signals are in good agreement with each other in terms of their peak eigenfrequency with a slight deviation. There is a slight mismatch in the locations of eigenfrequency between experimental and simulation results. It is challenging to optimize the experimental and simulation results exactly. We also show a unique multi-layered method of sample preparation where the majority of microparticles are lifted, floating, and almost lying in the same plane. The sample preparation method shown in this study can be instrumental for using microparticles for bio-sensing where the signals need to be distinguished. This study also establishes that different particles of different sizes can be identified as per their unique spectral signatures in the photoacoustic spectrum. As expected, the number of eigenfrequency increases with increase in the size of the microparticle; this might find application in colloidal sciences.

Multiphysics numerical modeling can be a promising tool for characterizing parameters responsible for the mechanism of laser-generated ultrasonics. This can be used efficiently to determine the best parameters for setting up the best experimental conditions. We show that information of the eigen vibration is contained in the photoacoustic signal using experimental demonstrations. The photoacoustic frequency spectrum reveals information about a particle such as eigenfrequency, which cannot be obtained from the time-domain signal only. On the other hand, the time-domain signal analysis allows us to calculate the velocities of different acoustic waves, for example, a Scholte wave based on its repetition after a certain interval of time.

Laser-generated acoustic waves inside polystyrene microparticles could be used as non-contact and in-situ measurements for additive manufacturing applications. They could also possibly be used in bio-sensing applications where microparticles can be coated with bio-samples or pathogens such as bacteria and measure the change in the thickness with the help of measurement in the shift of the time series obtained from coated and uncoated microparticles. The approach presented here can be extended to a class of anisotropic materials, to thin films with or without a substrate, and to layered solids. This simulation model can also be applied potentially further to single cells instead of spheres.

## Figures and Tables

**Figure 1 sensors-23-01776-f001:**
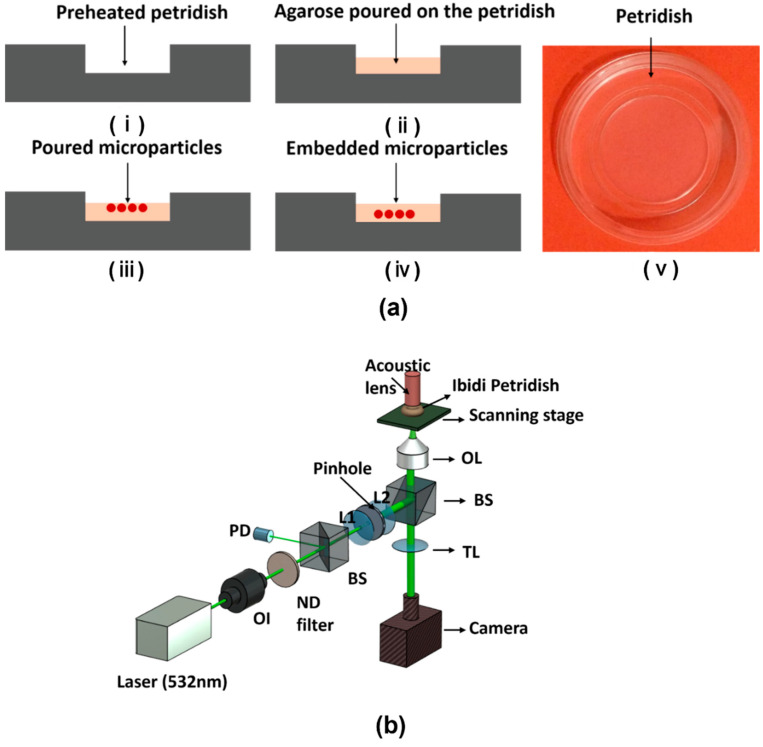
(**a**) Sample preparation steps for embedded sample step by step from (**i**–**v**). (**b**) Simplified PAM (Photoacoustic microscopy) experimental setup. OI: Optical isolator, BS: Beam splitter, ND filter: Neutral density filter, PD: Photodiode, OL: Objective lens. TL: Tube lens.

**Figure 2 sensors-23-01776-f002:**
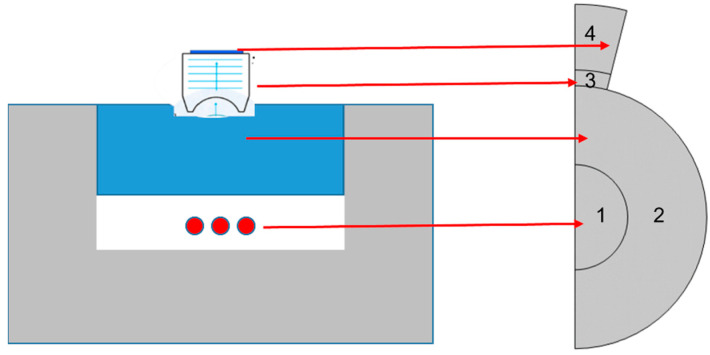
Geometry used for modelling in COMSOL (on the left) relating to the experimental setup. region 1: microparticles, region 2: water, region 3: piezoelectric region, region 4: backing layer.

**Figure 3 sensors-23-01776-f003:**
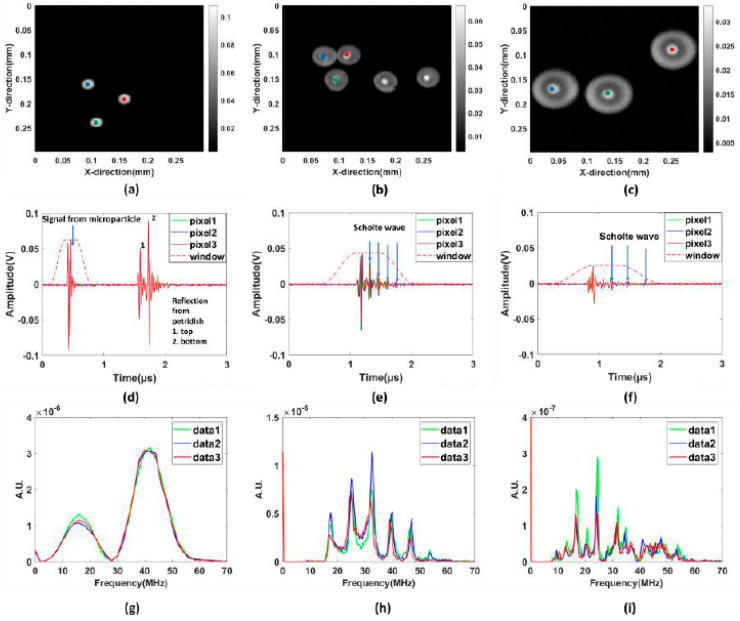
Experimental results for the photoacoustic imaging. Here the upper images show the maximum amplitudes obtained for the 3 particle sizes (**a**) PS20 (**b**) PS40 and (**c**) PS80 over the entire scanning area. Time series for pixels close to the centre of three particles in each image (marked with red, green, and blue dots) are shown with corresponding colours in (**d**–**f**) for PS20, PS40, and PS80, respectively. The amplitude spectra corresponding to the time series in (**d**–**f**) are shown in (**g**–**i**), respectively. For these spectra, the time series were multiplied with the windows shown in (**d**–**f**) to exclude reflections from the bottom of the Petri dish.

**Figure 4 sensors-23-01776-f004:**
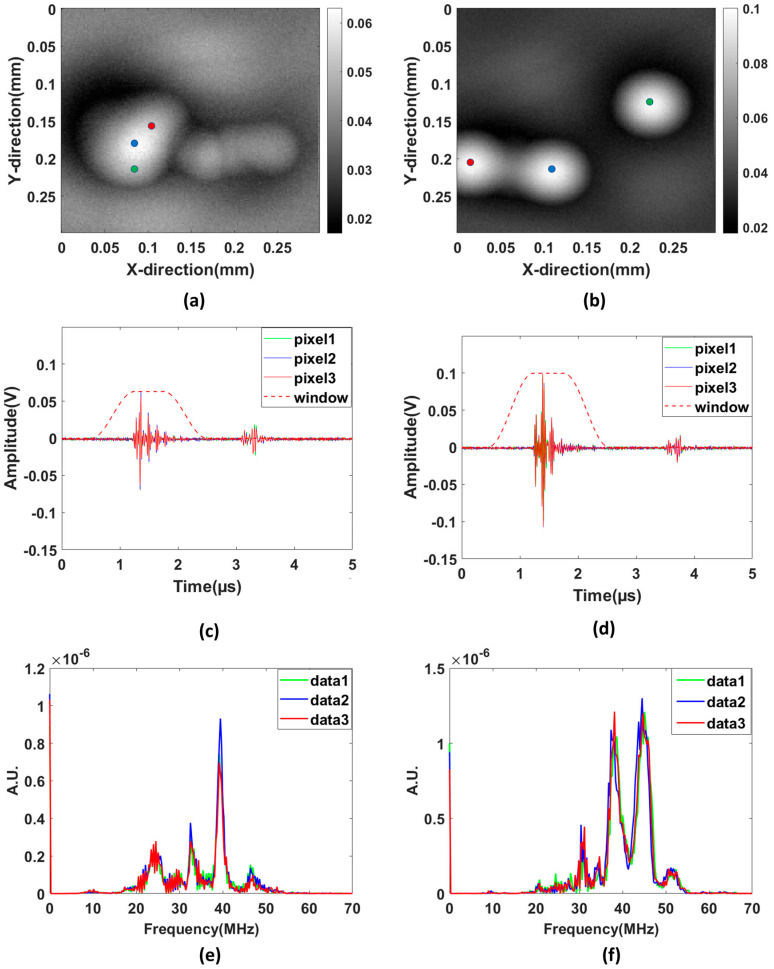
Experimental results for scanning acoustic imaging. Here, the upper images show the maximum amplitudes obtained for the 2 particle sizes (**a**) PS40 (**b**) PS80 over the entire scanning area. Time series for pixels close to the center of two particles in each image (marked with red, green, and blue dots) are shown with corresponding colors in (**c**,**d**) and for PS40 and PS80, respectively. The amplitude spectra corresponding to the time series in (**c**,**d**) are shown in (**e**,**f**) respectively.

**Figure 5 sensors-23-01776-f005:**
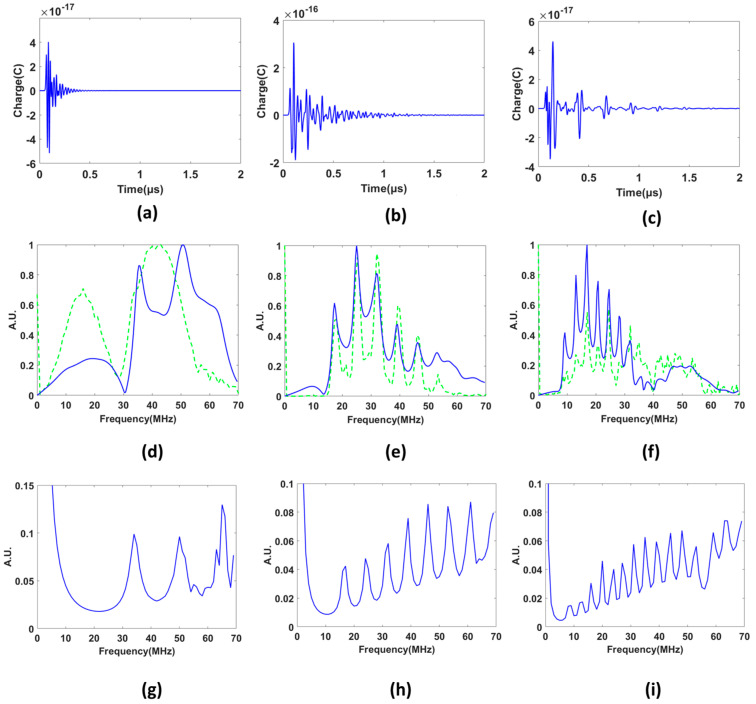
Simulated receiver signals for the laser excited (photoacoustic) pulses in the time-domain from the (**a**) PS20, (**b**) PS40, and (**c**) PS80 microparticles. The corresponding amplitude spectra for the COMSOL simulations are shown with solid blue lines in (**d**–**f**), respectively, with the experimental spectra for data 1 with the green stippled curves overlayed for comparison with simulated results in blue. In these figures, all spectra have been individual normalized with respect to their maximum values. The lower figures (**g**–**i**) shows the simulated frequency responses for PS20, PS40 and PS80, respectively.

**Figure 6 sensors-23-01776-f006:**
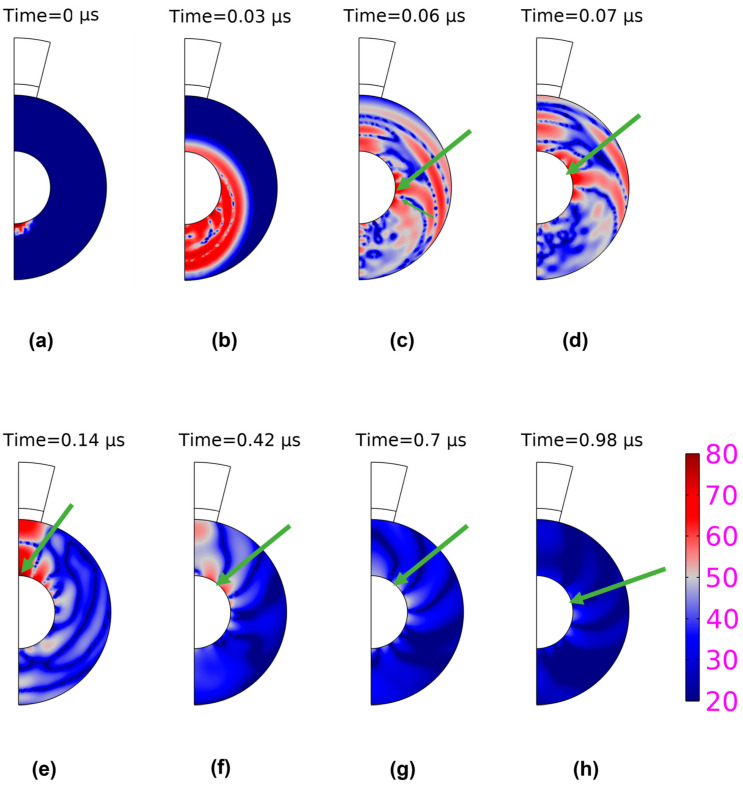
Spatial distributions of the pressure amplitude at eight different times generated from the numerical computation using a PS80 particle. The times are chosen to illustrate both the initial period with strong contribution from bulk waves (**a**–**c**) and remaining times (**d**–**h**) where Scholte waves becomes gradually more important. For figures where the Scholte wavelets could be identified, its centre is indicated with a green arrow. The time interval between the latter figures is chosen in accordance with the estimated wave period 0.28 µs for Scholte waves, to study the interface as it orbits several times around the microsphere. The used colormap similar for all figures shows the computed pressure in units Pascal a dB scale.

**Table 1 sensors-23-01776-t001:** Parameters for the microparticle, used in the COMSOL model.

Absorption coefficient	1/5 µm^−1^
Young’s modulus	3.8 × 10^9^ Pa
Poisson’s ratio	0.34
Density	1050 kg m^−3^
Thermal conductivity	0.035 (Wm^−1^K^−1^)
Spatial pulse width of laser beam	2.5 μm
Pulse duration of laser beam	1.25 ns
Laser intensity	1.45 × 10^5^ W cm^−2^
f-number of transducer	2
Backing layer thickness of transducer	50 μm
Piezoelectric thickness layer of transducer	12 μm

**Table 2 sensors-23-01776-t002:** (A) Predicted eigenfrequencies (in MHz) for different particle sizes based on simulations. (B) Experimentally obtained eigenfrequencies (in MHz) for different particle sizes.

Particle Size	f1	f2	f3	f4	f5	f6
**A**						
20 µm	35.7	50.7	--	--	--	--
40 µm	17.3	25.0	32.0	39.3	46.1	53.0
80 µm	20.0	24.5	28.0	31.9	35.7	44.6
**B**						
20 µm	33.0	--	--	--	--	--
40 µm	17.3	25.3	31.9	38.8	46.0	53.3
80 µm	20.6	24.6	28.0	32.0	35.0	45.0

## Data Availability

The data can be made available if needed.

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
