# Peer review of "Laser-Generated Scholte Waves in Floating Microparticles"

_sensors, 2023, doi:10.3390/s23041776_

Round 1
Reviewer 1 Report
In this manuscript, the authors study different kinds of laser-generated waves emitting inside the microparticles together, in which the Scholte waves are analyzed. Photoacoustic signals captured from the experiments show close agreement with those derived by the COMSOL simulation both in the time and frequency domain. The work is interesting and the manuscript is well organized, but there are major concerns to the authors’ claim.
1. For laser-generated waves emitted inside microparticles together, how to differentiate the different types of ultrasonic waves, and do they have their own characteristic frequencies, especially Scholte waves? Detaild explanations on the point would be helpful for the readers.
2. The authors conduct detailed theory and experiment on the power spectra analysis to the microspheres with varying sizes, as shown in Fig. 3 and Fig. 4. It would be helpful to indicate which spectrum is the Scholte wave?
3. Scholte waves are repeatedly generated in time domain according to the theory model based on COMSOL simulation. A more intuitive experimental demonstration on the Scholte waves should be offered.
4. The difference between Scholte waves and other waves should be indicated specifically in both theory and experiment.

Author Response
Thanks to the reviewer for very valuable comments! We understand the author’s concern regarding the claims and have therefore considered several methods for separating the fast-propagating bulk waves from the slower Scholte waves. Further details regarding these methods are given in the replies to the reviewer’s comments 1 to 4 under in the document attached.

Reviewer 2 Report
In this papers authors claims that they have been demonstrate the generation Scholte waves through the photoacoustic effect on microspheres suspended on agarose gel. To show their affirmation, they conducted a series of experiments on microspheres with three different diameters, namely 20, 40 and 80 micrometers, to reinforce they experimental approach, made element finite calculations with the COMSOL platform. Their idea is novel and could be revolutionary, however, it was not convincingly demonstrated. This is because the experimental eigenfrequencies does not match with the simulated ones. I guess, this could be due to the physical model used in the Finite Element calculations, this is oversimplified. For example:
1. the thermal confinement is no guaranteed with the microparticles size and the optical coefficient used (2x10^7 m^-1) in their simulations
2. The sensor model also is over simplified.
3. The results justification with dispersive waves is a misconception because this implies a frequency dependent sound speed, tha was not considered in their model.
In conclusion, the manuscript, as was presented, contains several significant issues so I am unable to recommend publication.
Minor corrections
There are a lot of typos, these were yellow-highlighted in the manuscript.

Author Response
Thanks to the reviewer for valuable comments and for pointing out errors in the manuscript!
Although some simplifications were needed in the FEM model e.g. to meet computational limitations, our general opinion is that the FEM calculations are not oversimplified as claimed by the reviewer. We believe that this can be verified by the quite close agreement with experimental and theoretical results if we for instance compare the peak locations for the Fourier spectrum also shown in table 2. Our opinion is also supported by reviewer 1 who said “Photoacoustic signals captured from the experiments show close agreement with those derived by the COMSOL simulation both in the time and frequency domain”.

Round 2
Reviewer 1 Report
The authors have addressed all my concerns, and the manuscript can be accepted in the journal.
Author Response
Thanks again to the reviewer for valuable feedback!
Reviewer 2 Report
See the attached file.

Author Response

(The authors gave the same response as above.)
